# Selecting the best machine learning algorithm to support the diagnosis of Non-Alcoholic Fatty Liver Disease: A meta learner study

**Paolo Sorino[1], Maria Gabriella Caruso[2], Giovanni Misciagna[3], Caterina Bonfiglio[1], Angelo Campanella[1], Antonella Mirizzi[1], Isabella Franco[1], Antonella Bianco[1], Claudia Buongiorno[1], Rosalba Liuzzi[1], Anna Maria Cisternino[4], Maria Notarnicola[2], Marisa Chiloiro[5], Giovanni Pascoschi[6]☯, Alberto Rubén Osella[1]☯*, MICOL Group¶**

1 Laboratory of Epidemiology and Biostatistics, National Institute of Gastroenterology, "S de Bellis" Research Hospital, Castellana Grotte, Bari, Italy, 2 Laboratory of Nutritional Biochemistry, National Institute of Gastroenterology, "S de Bellis" Research Hospital, Castellana Grotte, Bari, Italy, 3 Scientific and Ethical Committee, Polyclinic Hospital, University of Bari, Bari, Italy, 4 Clinical Nutrition Outpatient Clinic, National Institute of Gastroenterology, "S de Bellis" Research Hospital, Castellana Grotte, Bari, Italy, 5 San Giacomo Hospital Largo S. Veneziani, Monopoli, Bari, Italy, 6 Department of Electrical and Information Engineering, Polytechnic of Bari, Bari, Italy

☯ These authors contributed equally to this work.
¶ Membership of the MICOL Group is provided in the Acknowledgments.
* arosella@irccsdebellis.it

**Data Availability Statement:** All relevant data are within the manuscript and its Supporting Information files.

## Abstract

### Background & aims

Liver ultrasound scan (US) use in diagnosing Non-Alcoholic Fatty Liver Disease (NAFLD) causes costs and waiting lists overloads. We aimed to compare various Machine learning algorithms with a Meta learner approach to find the best of these as a predictor of NAFLD.

### Methods

The study included 2970 subjects, 2920 constituting the training set and 50, randomly selected, used in the test phase, performing cross-validation. The best predictors were combined to create three models: 1) FLI plus GLUCOSE plus SEX plus AGE, 2) AVI plus GLUCOSE plus GGT plus SEX plus AGE, 3) BRI plus GLUCOSE plus GGT plus SEX plus AGE. Eight machine learning algorithms were trained with the predictors of each of the three models created. For these algorithms, the percent accuracy, variance and percent weight were compared.

### Results

The SVM algorithm performed better with all models. Model 1 had 68% accuracy, with 1% variance and an algorithm weight of 27.35; Model 2 had 68% accuracy, with 1% variance and an algorithm weight of 33.62 and Model 3 had 77% accuracy, with 1% variance and an algorithm weight of 34.70. Model 2 was the most performing, composed of AVI plus GLUCOSE plus GGT plus SEX plus AGE, despite a lower percentage of accuracy.

**Funding:** This work was funded by a grant from the Ministry of Health, Italy (Progetto Finalizzato del Ministero della Salute, ICS 160.2/RF 2003), 2004/2006) and by Apulia Region-D.G.R. n. 1159, 28/6/2018 and 2019.

**Competing interests:** The authors have declared that no competing interests exist.

**Abbreviations:** ABSI, A Body Shape Index; AVI, Abdominal Volume Index; BAI, Body Adiposity Index; BIC, Bayesian Information Criteria; BMI, Body Mass Index; BP, Blood pressure; BRI, Body Roundness Index; Cp, Mallow's; DBP, Diastolic Blood Pressure; FLI, Fatty Liver Index; FN, False Negative; FP, False Positive; GGT, Gamma-Glutamyl Transferase; GOT Glucose, Glutamic-Oxaloacetic Transaminase; GPT, Glutamate Pyruvate Transaminase; HDL-C, High-Density Lipoproteins Cholesterol; HIS, Hepatic Steatosis Index; LDL-C, Low-Density Lipoproteins Cholesterol; ML, machine learning; MRI, magnetic resonance imaging; NAFLD, Non-Alcoholic Fatty Liver Disease; NASH, non-alcoholic steatohepatitis; $R^2$-Adj, R-squared Adjusted; RM, Regularized Multinomial; SBP, Systolic Blood Pressure; SVM, Support Vector Machine; TN, True Negative; TP, True Positive; US, ultrasound scan; WC, waist circumference; WHR, Waist-Hip Ratio; WHt.5R, Waist/Height$^{0.5}$; WHtR, Waist-to-Height Ratio.

## Conclusion

A Machine Learning approach can support NAFLD diagnosis and reduce health costs. The SVM algorithm is easy to apply and the necessary parameters are easily retrieved in databases.

## Introduction

Non-alcoholic fatty liver disease (NAFLD) is the leading cause of chronic liver disease in Western countries, as well as a condition raising the risk for cardiovascular diseases, type 2 diabetes mellitus and chronic renal disease, and increased mortality [1, 2].

Worldwide, NAFLD prevalence is currently estimated to be around 24% and is constantly increasing (from 15% in 2005 to 25% in 2010).

A meta-analysis published in 2016 reported an average prevalence of 23.71% in Europe [3]. Population-based studies conducted in our geographical area (district of Bari, Apulian Region, Italy), have estimated a NAFLD prevalence of around 30%, mainly among male subjects and elderly people [4].

NAFLD is defined as an accumulation of Triglycerides in hepatocytes (> at 5% of the liver volume) of a patient with reduced alcohol intake (<20 g / day in women or <30 g / day in men), after excluding viral infectious causes or other specific liver diseases [5].

NAFLD can manifest as pure fatty liver disease (hepato-steatosis) or as non-alcoholic steatohepatitis (NASH), an evolution of the former where the steatosis is associated with inflammation and hepatocellular injury, and with fibrogenic activation that can lead to cirrhosis and the onset of hepatocarcinoma [6].

According to recent EASL—EASD—EASO guidelines [7], at individual level the gold standard to identify steatosis in individual patients is magnetic resonance imaging (MRI), but ultrasound scanning (US) is preferred because it is more widely available and cheaper than MRI.

However, for large scale screening studies, serum bio markers and steatosis scores indexes have been preferred, as the availability and cost of US has a substantial impact on screening feasibility.[7] One of the best validated INDEXES is the Fatty Liver Index (FLI) [8], although other scores or anthropometric measures work as well as the FLI in predicting the NAFLD risk [9].

The FLI is important and has widespread use in the diagnosis of NAFLD in epidemiological studies because it relies on few parameters that are easy to obtain in clinical practice.

In a previous study, we compared the performance of eight alternative hybrid methods to identify subjects with NAFLD in a large population [9]. In that setting, predictive formulas were used as a first approach strategy, to identify subjects considered to be at greater risk of steatosis, needing to be scheduled for later US. In clinical practice, it is necessary to develop support models for the diagnosis of NAFLD that include only clinical and laboratory routine parameters which are easily retrievable from health databases.

In recent years, and due to the increasing prevalence of NAFLD, there has been a research trend towards low-cost diagnostic methods, and Machine Learning has been identified as a valid tool. Machine Learning (ML) is a branch of artificial intelligence that aims to allow machines to work using intelligent "learning" software [10]. Data sets are provided, that the machine processes through algorithms which develop their own logic to perform the function, the activity or the task required. Machine Learning has already been used as a support tool for

the diagnosis of some diseases and for risk quantification, such as Cardiovascular risk in patients with Diabetes Mellitus [11, 12], Ischemic Heart Disease [13] and Cancers [14].

The purpose of our study is to compare various Machine Learning algorithms, using easily available laboratory parameters, in order to find the best algorithm to support the identification of subjects at greater risk of NAFLD to be scheduled for US assessment. We also compared the performances of the trained models; to assess the true reliability and ease of use of each, and validated the results on a large population-based study.

## Materials and methods

### Population

Details about the study population have been published elsewhere [9]. Briefly, a prospective cohort study was conducted by the Laboratory of Epidemiology and Biostatistics of the National Institute of Gastroenterology, "Saverio de Bellis" Research Hospital (Castellana Grotte, Bari, Italy). The MICOL Study is a population-based prospective cohort study randomly drawn from the electoral list of Castellana Grotte (≥30 years old) in 1985 and followed up in 1992, 2005–2006 and 2013–2016. In 2005–2006 this cohort was added with a random sample of subjects (PANEL Study) aged 30–50 years, to compensate for the cohort aging. In this paper the baseline for the MICOL cohort was established in 2005–2006 to capture all ages.

Participants were interviewed to collect information about sociodemographic characteristics, health status, personal history and lifestyle factors. Weight was taken with the subject in underwear, on a SECA® electronic balance, and approximated to the nearest 0.1 kg. Height was measured with a wall-mounted stadiometer SECA®, approximated to 1 cm. Blood pressure (BP) measurements were performed following international guidelines [15]. The mean of 3 BP measurements was calculated.

A fasting venous blood sample was drawn, and the serum was separated into two different aliquots. One aliquot was immediately stored at −80˚C. The second aliquot was used to test biochemical serum markers by standard laboratory techniques in our Central laboratory.

The study included a total of 2970 out of 3000 selected subjects; 56.5% were male. In 1985, 2472 subjects had been enrolled. The cohort was followed-up in 1992 and, in 2005–2006, 1697 from the original cohort were still present. A further sample of 1273 subjects were enrolled in 2005–2006 to compensate for the cohort aging [16]. All subjects gave informed written consent to participate.

All procedures were performed in accordance with the ethical standards of the institutional research committee (IRCCS Saverio de Bellis Research and with the 1964 Helsinki declaration, and had had Ethical Committee approval for the MICOL Study (DDG-CE-347/1984; DDG-CE-453/1991; DDG-CE-589/2004; DDG-CE 782/2013).

## Model development

### The decisive variables and the training sessions of the models

After the adoption of the Best Subset Selection [17], we compared three methods to choose the optimal model: Mallow's C$p$ (Cp) [18], Bayesian Information Criteria (BIC) [19] and, R-squared Adjusted (R$^2$Adj) [20].

The following 27 variables were considered when employing the three subset selection methods: Age, Sex, Height, Body Mass Index (BMI) [21], Waist Circumference (WC) [22], Systolic Blood Pressure (SBP), Diastolic Blood Pressure (DBP), A Body Shape Index (ABSI) [23], Abdominal Volume Index (AVI) [24], Body Adiposity Index (BAI) [25], Body Roundness Index (BRI) [26], Hepatic Steatosis Index (HIS) [27], Leukocyte Alkaline Phosphatase Score

[28], Total Bilirubin, Alkaline Phosphatase, Glucose, Gamma-Glutamyl Transferase (GGT), Glutamic-Oxaloacetic Transaminase (GOT), Glutamate Pyruvate Transaminase (GPT), High-Density Lipoproteins Cholesterol (HDL-C), Low-Density Lipoproteins Cholesterol (LDL-C), Triglycerides, FLI, Waist-Hip Ratio (WHR) [22], Waist-to-Height Ratio (WHtR) [22], Waist/Height$^{0.5}$ (WHt.5R) [29] and Diabetes. All biochemical markers were introduced in the models as continuous variables. We choose to introduce the biochemical markers as continuous variables to better reflect the natural scale of the variables. The assumption behind this choice is that the effect of categorization is the loss of information.

The following best subset results were available after running the code. Cp: Age, Sex, AVI, BRI, FLI, GGT, Glucose, GPT, HIS, PAD, WHR, WHtR, WHt.5R, WC; BIC: Age, Sex, AVI, BRI, FLI, GGT, Glucose and R$^2$-Adj: Age, Sex, AVI, BRI, FLI, GGT, Glucose, GPT, HIS, PAD, PAS, Triglycerides, WHR, WHtR, WHt.5R and WC. The best model was then extrapolated from the BIC method, because the trade-off between number of variables and explained variation did not show substantial changes. The model was composed with one of the following indexes: FLI, AVI or BRI plus Age, Sex, Glucose and Gamma-glutamyl transpeptidase (GGT).

Table 1 shows the formulas employed in this study and the variables necessary for their construction.

These formulas are simple and the variables (essential to calculate them) are easily available because they are collected and routinely assessed as anthropometric measures and biochemical markers. The selected database included all the variables essential for the construction of the predictive formulas, along with other values such as: Age, Sex, GGT and Glucose. However, GGT was not included in the model employing the FLI Index formula because it is already included in the formula.

Following the European Association for the Study of the Liver (EASL), European Association for the Study of Diabetes (EASD) and European Association for the Study of Obesity (EASO) recommendations [7], NAFLD diagnosis was performed using an ultrasound scanner Hitachi H21 Vision (Hitachi Medical Corporation, Tokyo, Japan). Examination of the visible liver parenchyma was performed with a 3.5 MHz transducer.

## Model implementation

The algorithms considered were the following:

- Boosting Tree Classifier (using AdaboostClassifier) [30, 31]

- Decision Tree Classifier [32]

- Naive Bayes Classifier [33, 34]

- K-Nearest Neighbors Classifier [35]

**Table 1. Indexes formula and their structure.**

| Reference | Name | Formula |
|---|---|---|
| Bedogni G [8] | Fatty Liver Index (FLI) | $FLI = \frac{e^{0.953*\ln(TG)+0.139*BMI+0.718*\ln(GGT)+0.053*WC-15.745}}{1+e^{0.953*\ln(TG)+0.139*BMI+0.718*\ln(GGT)+0.053*WC-15.754}} * 100$ |
| Guerrero-Romero F [24] | Abdominal Volume Index (AVI) | $AVI = \frac{\left[2*(WC)^2+0,7*\left(\frac{WC}{HC}\right)^2\right]}{1000}$ |
| Thomas DM [26] | Body Roundness Index (BRI) | $BRI = 364.2 - 365.5\sqrt{1 - \left(\frac{\left(\frac{WC[m]}{2\pi}\right)^2}{(0.5*Ht[m])^2}\right)}$ |

**Abbreviation** TG = Triglycerides; BMI = Body Mass Index; GGT = Gamma-Glutamyl Transferase; WC = Waist Circumference; HC = Hip Circumference; Ht = Height (centimeters).

- Neural Network Classifier [36]

- Random Forest Classifier [37]

- Regularized Multinomial Classifier (use Logistic regression) [38, 39]

- Support Vector Machine Classifier [40, 41]

All eight algorithms considered were supervised learning models, except for the neural network algorithms, which consist of reinforcement learning. Afterwards, a code was developed and due to the interaction with Python, it recalled the functions of the Machine Learning contained in the Scikit-learn [42] library. This was used to compare all models in order to identify the one best suited to diagnose NAFLD, and also the model minimizing false-negative predictive values.

## The actual training sessions

The models considered were the following:

2.1. FLI plus Glucose plus Age plus Sex

2.2. AVI plus Glucose plus GGT plus Age plus Sex

2.3. BRI plus Glucose plus GGT plus Age plus Sex

For each model above, all the algorithms were compared using the Meta-learner approach [43, 44]

Initially, it was important to pre-process data to eliminate missing values for features (x) or for target variables (y). Then, we obtained a dataset containing 2868 subject Subsequently, we set the algorithm to randomly extract 50 individuals from the initial database, which were used for the construction of a new dataset. This dataset was used for the prediction phase, making use of machine learning algorithms that had been adequately trained during the training phase. During the training sessions, all 2868 subject was used for the training and for the test by means of the 10-fold cross-validation approach [45]. To retrieve the best parameters for each algorithm, GridSearchCV, which is a method contained in a Scikit-learn in Python, was used. In this way, the algorithm characterized by the lowest variance was identified, to reduce the possibility of a false prediction. After selection of the best parameters for each algorithm, we carried out the prediction on a dataset of 50 individuals (generated by the prediction), thus obtaining the predicted target variable for each algorithm.

## The analysis of model performance

We compared eight different types of Machine Learning algorithms, in terms of the percentage accuracy and variance for each model. The parameters used for comparison in this case were: values of accuracy, variance, calculated confidence limits (95%), the weight of each model (as a %) and the number of ultrasound examinations it could avoid.

The accuracy of a model is defined as:

$$Accuracy = \frac{Number\ of\ correct\ predictions}{Total\ number\ of\ predictions} * 100 \ [46]$$

More specifically, the accuracy of a model is calculated with the following formula:

$$Accuracy = \frac{TP+TN}{TP+TN+FP+FN} \ [46]$$

Where TP = True Positive, TN = True Negative, FP = False Positive and FN = False Negative.

Another aspect to take into account was the value of variance for each algorithm, as a key indicator of the variability of a dataset.

**Table 2. Subset subject characteristics by NAFLD condition.** MICOL Study, Castellana Grotte (BA), Italy, 2005.

| Variables | NAFLD | |
|---|---|---|
| | Absent | Present |
| N (%) | 2033 (68.45) | 937(31.55) |
| Sex | | |
| Female | 1007 (49.5) | 286 (30.5) |
| Male | 1026 (50.5) | 651 (69.5) |
| Age | 54.04 (15,61) | 55.42 (13,67) |
| FLI | 31,09 (25,50) | 64,11 (24,47) |
| AVI | 15,96 (4,27) | 21,15 (4,96) |
| BRI | 4,48 (1,66) | 6,21 (1,89) |
| GLUCOSE | 105,51 (24,03) | 117,62 (33,13) |
| GGT | 14,60 (15,18) | 20,57 (19,75) |

Cells reporting subject characteristics contain mean (±SD) or n (%).

A low value of variance means that the predictions have a percentage of accuracy with a low variability. Differently a high value of variance indicates a weak prediction also in case of high value of accuracy percentage.

Another fundamental parameter we considered was the weight of the algorithms, i.e. the value indicating the algorithm with the best performance in the test phase.

Of 100%, each algorithm had a weight percentage, and the one that contributed most to the sum of all the weights was taken as the best algorithm. In addition, for each algorithm under study, the optimal parameters for the future development of an algorithm with the absolute highest values of accuracy and weight were calculated.

## Results

Subjects characteristics and the performance of each of the three indexes considered are shown in Table 2. NAFLD prevalence was 31.55% and was, as expected, more prevalent among men. Subjects with NAFLD were a little older, and had increased levels of Glucose and GGT.

### FLI plus glucose plus age plus sex predictive model

In the training session of the Model employing FLI plus Glucose plus Age plus Sex as predictive variables, the Boosting Tree was considered as the algorithm with greatest accuracy, characterized by 76% accuracy and by a weight factor of 19.02% while the algorithm with the lowest variance was the Support Vector Machine with 68% accuracy, lower than the Boosting Tree, but the weight of the model was 27.35%. Compared to the Boosting Tree, the Support Vector Machine made less mistakes during the testing phase.

In Fig 1 we show the trend of the error in the test phase and the trend of the error in the training phase for each model.

### AVI plus glucose plus GGT plus age plus sex predictive model

In the training session of the Model employing AVI plus Glucose plus GGT plus Age plus Sex as the predictive variables, the Regularized Multinomial (RM) was considered as the algorithm with the greatest accuracy, characterized by 76% accuracy and by a weight factor of 28.32%.

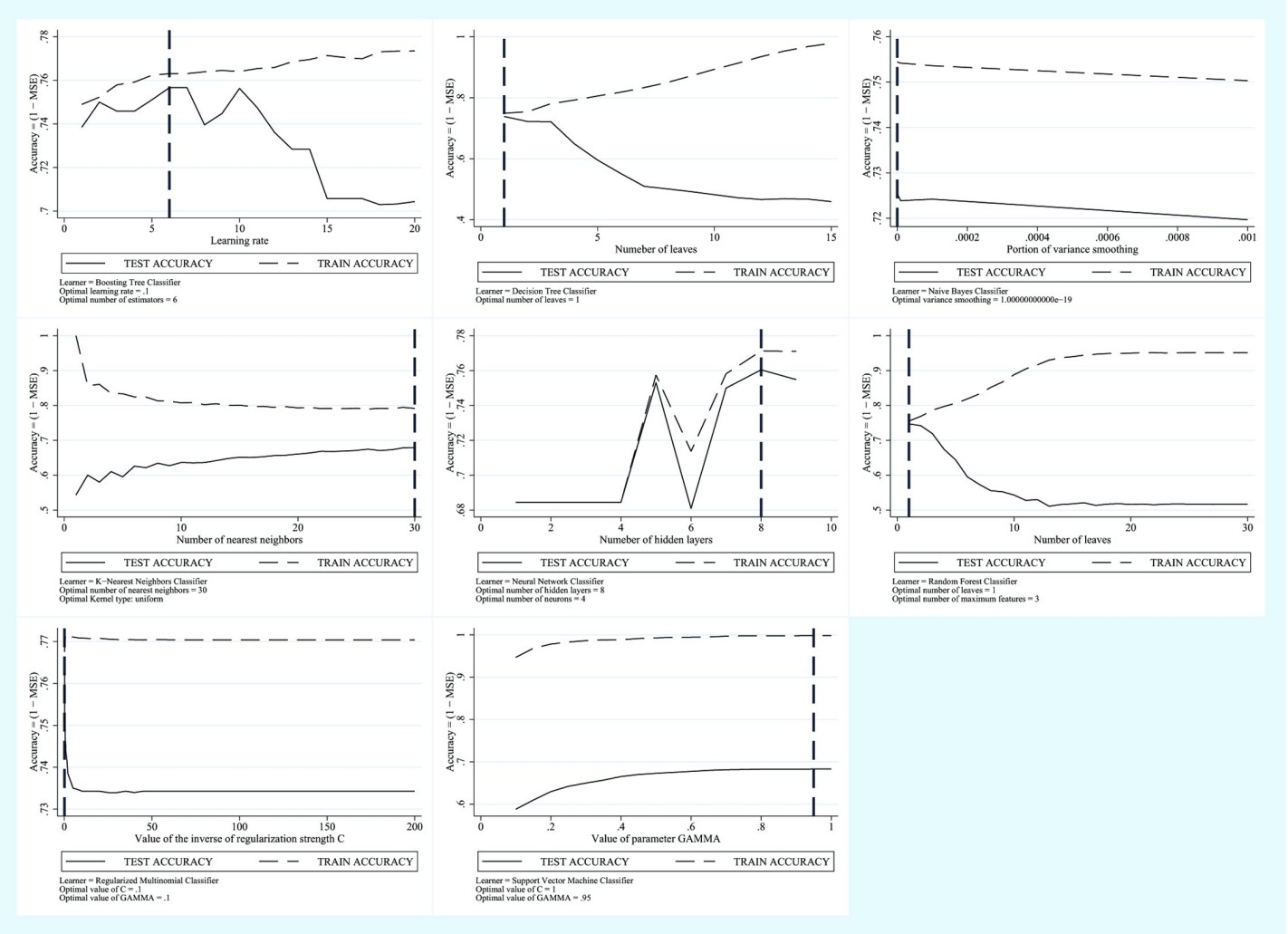

**Fig 1. Machine learning applied to NAFLD diagnosis.** Training Error vs Test Error for the FLI plus Glucose plus Age plus Sex Predictive Model.

while the algorithm with the lowest variance was the Support Vector Machine, with 68% accuracy, lower than the RM but the weight of the algorithm was 32.62%.

Compared to the RM, the Support Vector Machine made less mistakes during the testing phase.

In Fig 2 we show the trend of the error in the test phase and the trend of the error in the training phase for each model.

## BRI plus glucose plus GGT plus age plus sex predictive model

In the training session of the Model employing BRI plus Glucose plus GGT plus Age plus Sex as the predictive variables, the Nearest Neighbor was considered as the algorithm with the greatest accuracy, characterized by 77% accuracy and by a weight factor of 22.21%, while the algorithm with the lowest variance was the Support Vector Machine, with 77% accuracy, equivalent to the Regularized Multinomial RM, but the model weight was 34.70%.

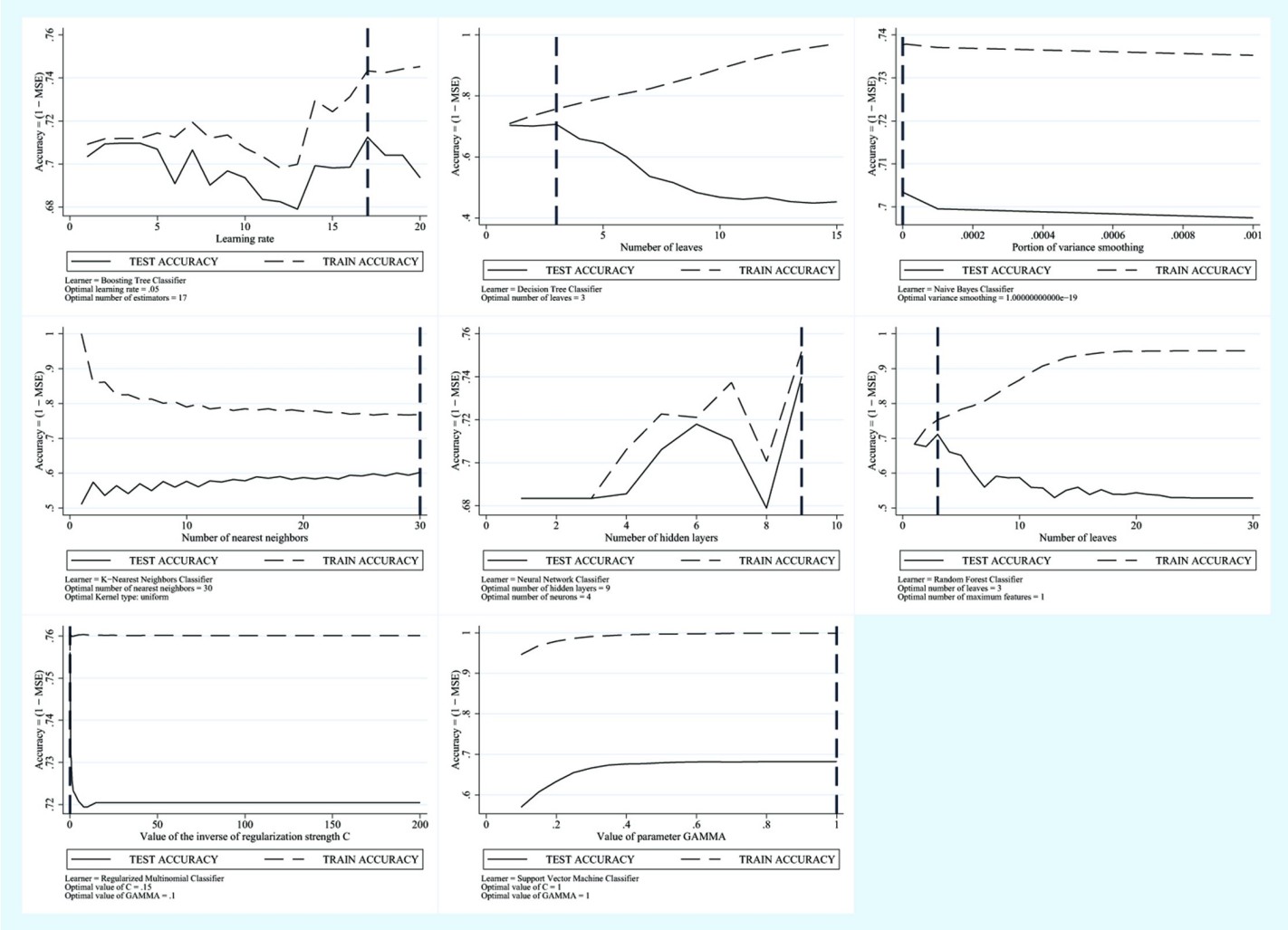

**Fig 2. Machine learning applied to NAFLD diagnosis.** Training Error vs Test Error for the AVI plus Glucose plus GGT plus Age plus Sex Predictive Model.

Despite having the same percentage of accuracy as the RM, the Support Vector Machine made less mistakes during the testing phase.

In Fig 3 we show the trend of the error in the test phase and the trend of the error in the training phase for each model.

Figs 4, 5 and 6 show the results, in terms of accuracy, of the test with 95% confidence intervals, the weight of the model and also the value of the variance.

From the above Figures, it can be seen that the SVM algorithm, even if it did not feature the highest accuracy percentage, was the best for predicting NAFLD.

From the output of the three algorithms, we have also calculated the value of sensibility, specificity, predictive value of a positive and negative test in the prediction phase and are shown in Table 3.

## Discussion

In this study, the search for the best algorithm to support NAFLD diagnosis was conducted by comparing the different Machine Learning algorithms for each model.

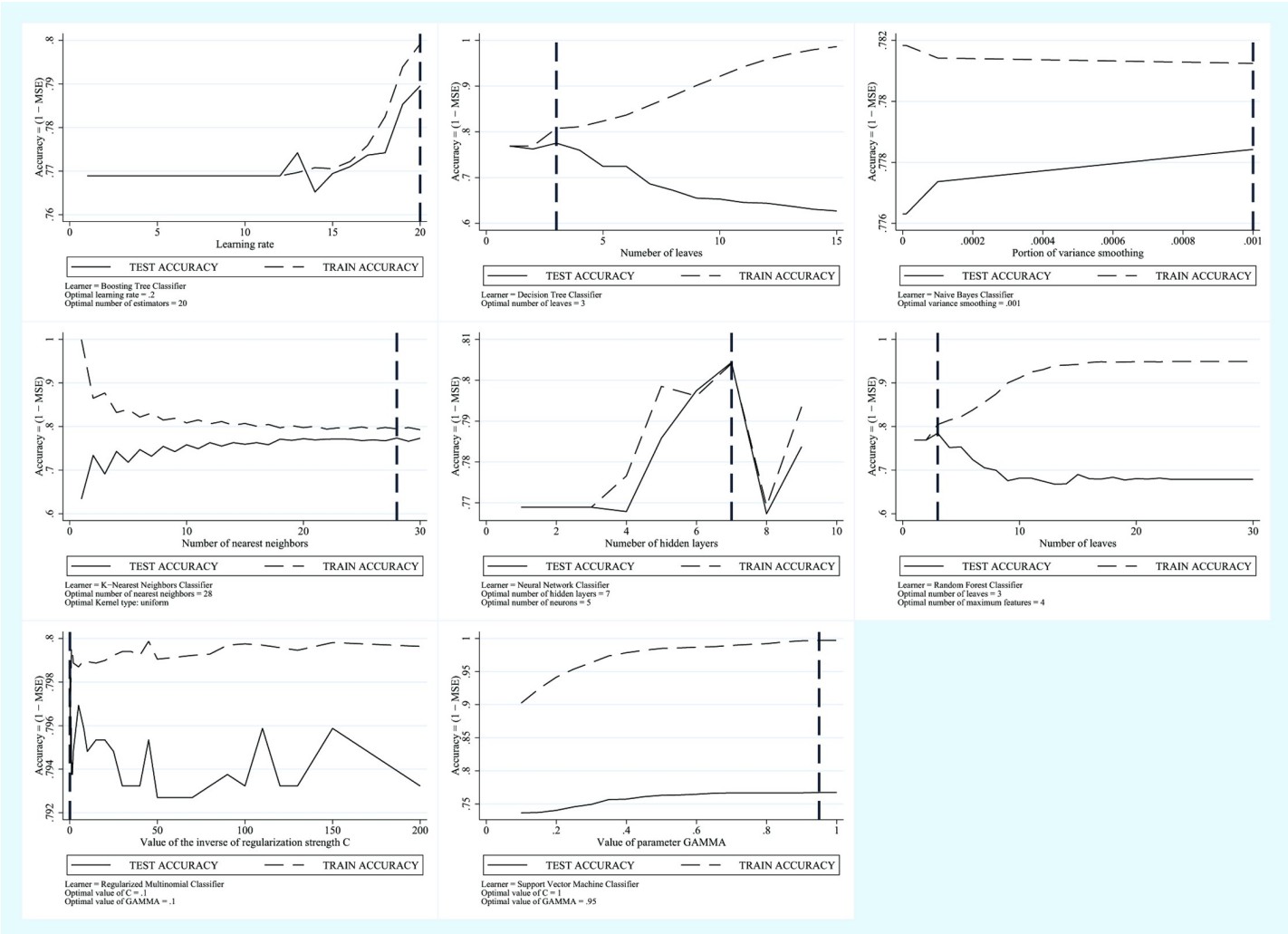

**Fig 3. Machine Learning Applied to NAFLD Diagnosis.** Training Error vs Test Error for the BRI plus Glucose plus GGT plus Age plus Sex Predictive Model.

Specifically, the model with a high level of accuracy and the model with the lowest level of variance were identified in this research. The Machine Learning model presenting the lowest variance was selected.

Today, the search for non-invasive methods is very important, considered as an alternative to the expensive NAFLD diagnostic tools (MRI, Ultrasound). The reorganization of the National Health System requires closer consideration of aspects linked to the performance together with the factors linked to cost-reduction and waiting times. The aim of our study was to use new, modern Machine Learning techniques to support medical decisions during the diagnostic phase using easier and cheaper tools, thus reducing both the costs and waiting times inherent to the use of instrumental methods.

In view of the results of this study, it is possible to state that the most appropriate Machine Learning Algorithm is the Support Vector Machine in Python. In particular, the Support Vector Machine employing AVI plus Glucose plus GGT plus Sex plus Age, despite having an almost identical percentage of accuracy and weight to the other models, produced fewer prediction errors in the test step. We obtained in the test phase for the models composed of FLI

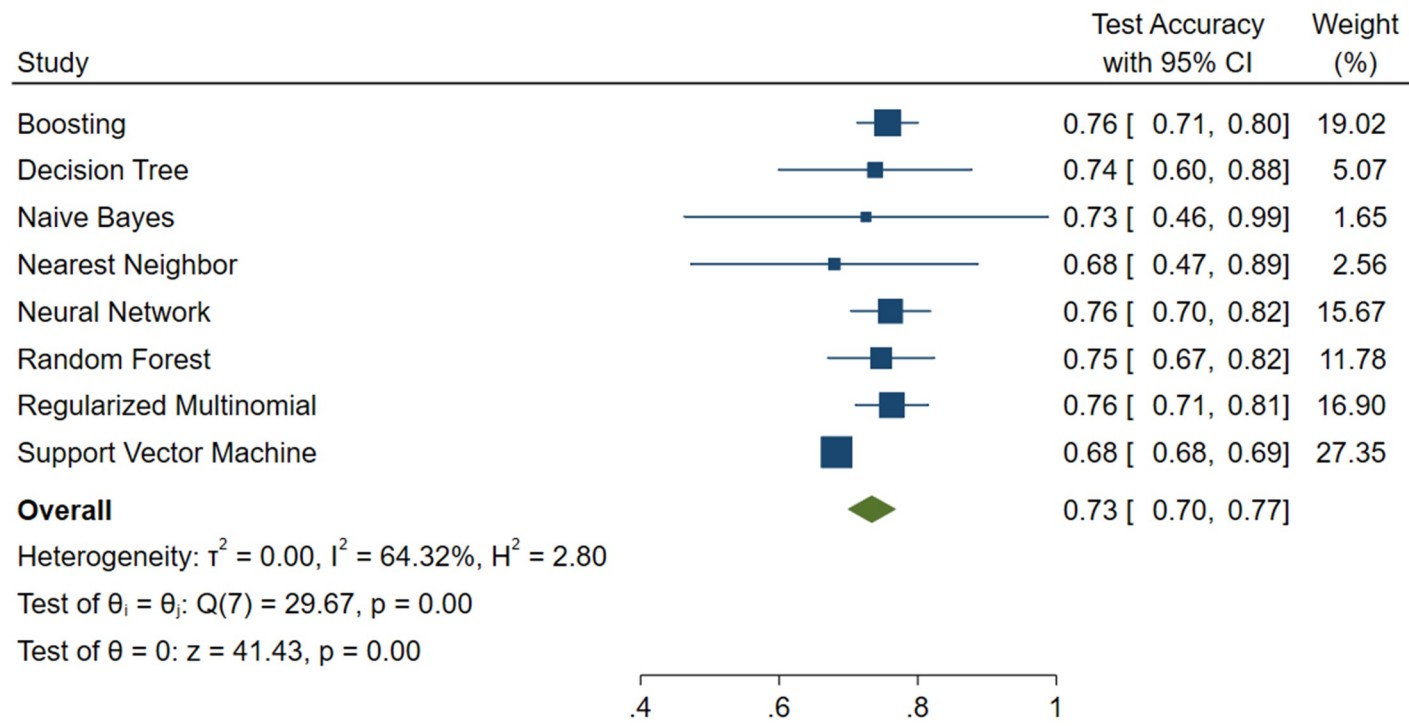

**Fig 4. Machine learning applied to NAFLD diagnosis.** Forest Plot for the FLI plus Glucose plus Age plus Sex Predictive Model.

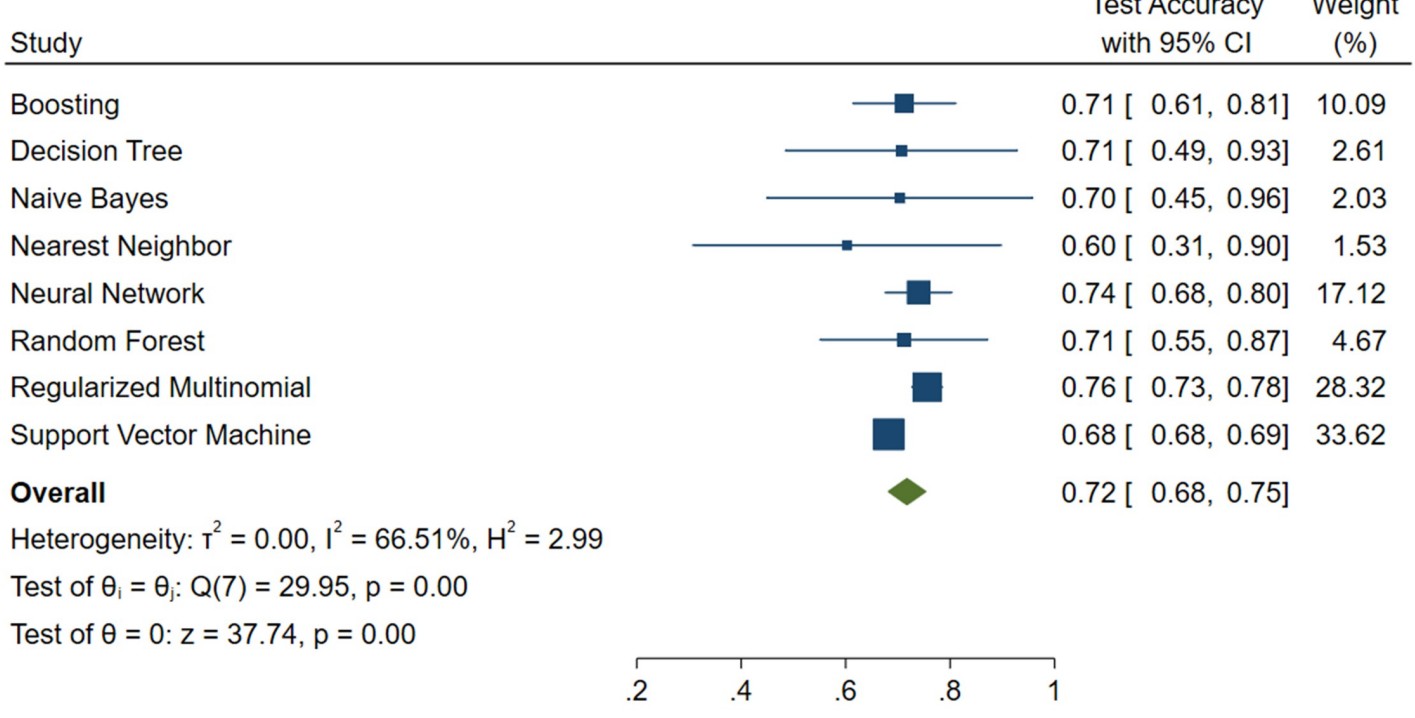

**Fig 5. Machine learning applied to NAFLD diagnosis.** Forest Plot for the AVI plus Glucose plus GGT plus Age plus Sex Predictive Model.

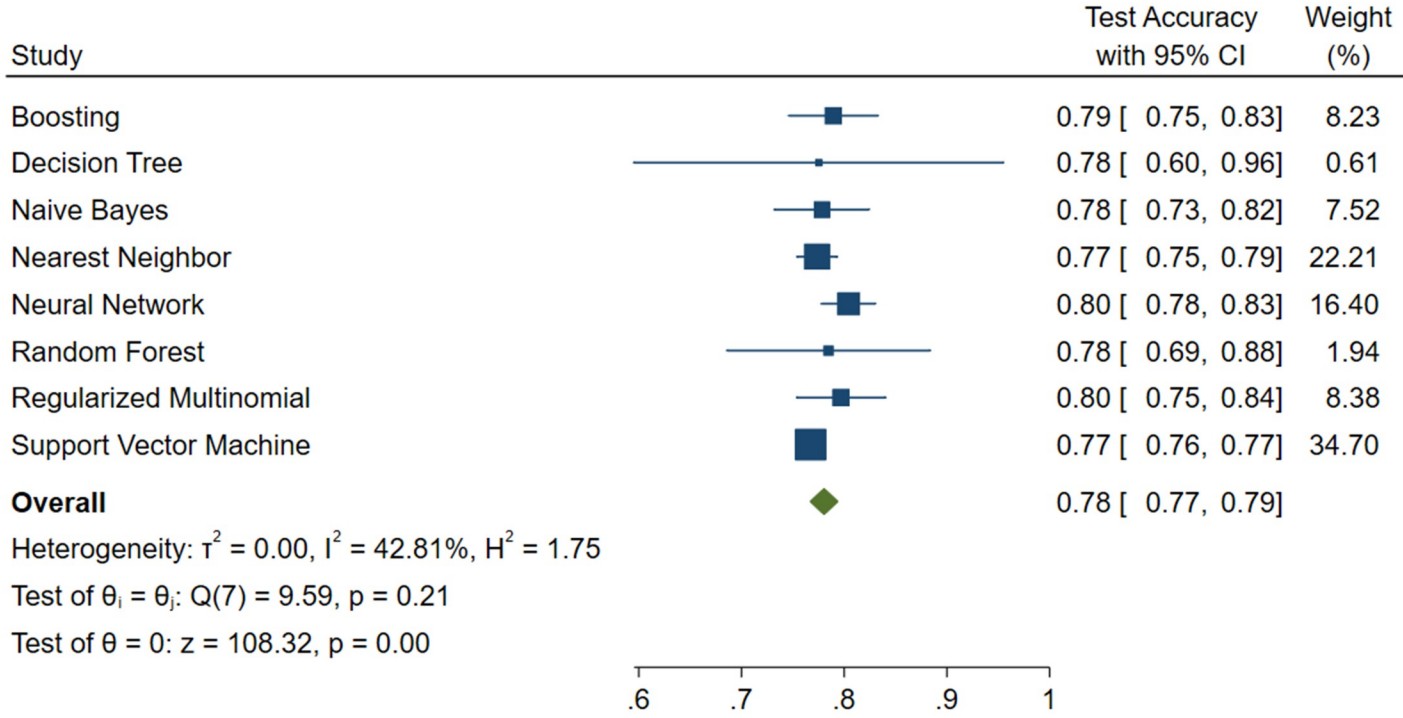

**Fig 6. Machine learning applied to NAFLD diagnosis.** Forest Plot for the BRI plus Glucose plus GGT plus Age plus Sex Predictive Model.

plus Glucose plus Age plus Sex and AVI plus Glucose plus GGT plus Age plus Sex a percentage error equal to 32% while for the model composed of BRI plus Glucose plus GGT plus Age plus Sex an error of 23%. However, in the prediction phase, the model that made fewer errors was the one composed of AVI plus Glucose plus GGT plus Age plus Sex with an error of 20% while FLI plus Glucose plus Age plus Sex 26% and BRI plus Glucose plus GGT plus Age plus Sex 28%.

Therefore, AVI plus Glucose plus GGT plus Sex plus Age was the model that contributed most to reducing unnecessary ultrasound examinations.

The good performance of ML Algorithms used to identify NAFLD, applying common anthropometric parameters and other variables, has shown them to be a valid alternative to classic Indexes [47, 48].

Moreover, the SVM was well able to identify subjects without NAFLD. From an ethical perspective, the model with the lowest variance is the best one, as it is characterized by a smaller number of false negatives, despite a lower percentage of accuracy.

**Table 3.**

|  | Sensibility | Specificity | VPP | VPN |
|---|---|---|---|---|
| FLI plus Glucose plus Age plus Sex | 0.979 | 1.00 | 1.00 | 0.990 |
| AVI plus Glucose plus GGT plus Age plus Sex | 0.985 | 1.00 | 1.00 | 0.993 |
| BRI plus Glucose plus GGT plus Age plus Sex | 0.967 | 0.99 | 0.997 | 0.990 |

This kind of study underlines the fact that this ML Algorithm can be used to find subjects at high risk of NAFLD, who need to undergo US. Furthermore, as low-risk subjects do not undergo US, 81.9% of unnecessary US examinations could be avoided (this value was calculated as the ratio of the total number of subjects in the test set divided by the total number of subjects in the test set plus the number of incorrect predictions.)

Some methodological issues need to be considered. A strength of this study is the population-based random sample from which the observations were drawn. The NAFLD prevalence in the sample is a good estimator of the population prevalence and its age-sex distribution. Limitations include both the limited number of observations and the method used to perform NAFLD diagnosis. It may be criticized the low sensibility of the NAFLD diagnostic methodology, as it fails to detect fatty liver content >25–90% [49]. However, this is a population-based study and subjects were chosen from the electoral register. They did not seek medical attention and participated on volunteer basis. Then, the diagnosis of NAFLD performed by US was the only diagnostic procedure we could propose to the participants. Ethical issues prevent us to propose biopsy or H-MRS. Moreover, and to lighten the waiting lists, our purpose was to find out a machine learning algorithm that permit us to avoid a number of USs which otherwise would have been prescribed. Then, this algorithm is useful to exclude NAFLD and as valid support diagnostic in the context of epidemiologic studies and not as replacement diagnostic tool.

In conclusion, this model, like others based on ML Algorithms, may be considered as a valid support for medical decision making as regards health policies, in epidemiological studies and screening.

## Supporting information

**S1 File. Dataset with random selection.**
(XLS)

## Acknowledgments

Computations for this research were performed in the Laboratory of Epidemiology and Biostatistics, National Institute of Gastroenterology, "S de Bellis" Research Hospital, MICOL Working Group: Vittorio Pugliese (Laboratory of Epidemiology and Biostatistics), Mario Correale, Palma Iacovazzi, Anna Mastrosimini, Giampiero De Michele (Laboratory of Clinical Pathology), Osvaldo Burattini (Unit of Gastroenterology), Valeria Tutino, Benedetta D'Attoma (Laboratory of Nutritional Biochemistry), Maria R Noviello (Department of Radiology), National Institute of Gastroenterology "S de Bellis" Research Hospital, Castellana Grotte (BA), Italy.

## Author Contributions

**Conceptualization:** Paolo Sorino, Giovanni Pascoschi, Alberto Rubén Osella.

**Data curation:** Paolo Sorino, Caterina Bonfiglio, Angelo Campanella, Antonella Mirizzi, Isabella Franco, Antonella Bianco, Claudia Buongiorno, Rosalba Liuzzi, Anna Maria Cisternino, Maria Notarnicola, Alberto Rubén Osella.

**Formal analysis:** Paolo Sorino.

**Investigation:** Paolo Sorino, Marisa Chiloiro, Alberto Rubén Osella.

**Methodology:** Paolo Sorino, Marisa Chiloiro.

**Project administration:** Alberto Rubén Osella.

**Supervision:** Giovanni Pascoschi, Alberto Rubén Osella.

**Writing – original draft:** Paolo Sorino, Maria Gabriella Caruso, Giovanni Misciagna.

**Writing – review & editing:** Giovanni Pascoschi, Alberto Rubén Osella.

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
