## [Decision Letter · Decision Letter 0]

7 Aug 2020

PONE-D-20-19706

SELECTING THE BEST MACHINE LEARNING ALGORITHM TO SUPPORT THE DIAGNOSIS OF NON-ALCOHOLIC FATTY LIVER DISEASE: A META LEARNER STUDY.

PLOS ONE

Dear Dr. Osella,

Thank you for submitting your manuscript to PLOS ONE. After careful consideration, we feel that it has merit but does not fully meet PLOS ONE’s publication criteria as it currently stands. Therefore, we invite you to submit a revised version of the manuscript that addresses the points raised during the review process.

Especially, please expand the sample size of validation cohort.

We look forward to receiving your revised manuscript.

Kind regards,

Ming-Lung Yu, MD, PhD

Academic Editor

PLOS ONE

Journal Requirements:

2. Thank you for including your ethics statement:  "Ethical Committee approval for the MICOL Study (DDG-CE-347/1984; DDG-CE-453/1991; DDG-CE-589/2004; DDG-CE 782/2013".   

"This work was funded by a grant from the Ministry of Health, Italy (Progetto Finalizzato

del Ministero della Salute, ICS 160.2/RF 2003), 2004/2006) and by Apulia Region-D.G.R. n. 1159,

28/6/2018 and 2019"

Reviewers' comments:

Reviewer's Responses to Questions

**Comments to the Author**

1. Is the manuscript technically sound, and do the data support the conclusions?

Reviewer #1: Yes

Reviewer #2: Yes

2. Has the statistical analysis been performed appropriately and rigorously? 

Reviewer #1: Yes

Reviewer #2: Yes

3. Have the authors made all data underlying the findings in their manuscript fully available?

Reviewer #1: Yes

Reviewer #2: No

4. Is the manuscript presented in an intelligible fashion and written in standard English?

Reviewer #1: Yes

Reviewer #2: Yes

5. Review Comments to the Author

Reviewer #1: Authors did a tremendous effort building the database and analyzing different AI algorithms to compare them, but I think that the networks they have built are unbalanced and some other methodological concerns that they should address.

• The authors did make a database including 2970 patients. They used 2920 cases for the training set and only 50 to test the algorithm. From the point of view of this reviewer these 50 represent less 2% of them, but it is recommended to include at least 15% of the data for validation purpose. Moreover, cases are well represented when they are selected randomly, as they did.

• Authors did use accuracy and variance to determine if the algorithm works. However, usually sensitivity and specificity are calculated because accuracy, although it takes into account the FP-FN, does not distinguish them separately, and it is as important that your algorithm is correct as it is not wrong, especially when it comes to the diagnosis of a disease.

• As for the description of the population they have used, there are several concerns. On the one hand, from the total cases, only about 32% have NAFLD, which makes the network not compensated and, be able to detect better the cases that do not have NAFLD than those that have NAFLD (something that they mention in the Discussion part, "the SVM was well able to identify subjects without NAFLD", obviously, because it has been trained with more cases without NAFLD). This can lead to very good accuracy results if the test cases are selected randomly and none of them is a NAFLD patient (and in their case it can happen since few test cases have been used). On the other hand, the network is also gender imbalanced, 56.5% male. Perhaps it would have been better to create two different networks, one for men and another for women. This decompensation is even greater in the case of women with NAFLD, there are only 30.5% of the cases, and they do not mention at any moment, neither in the conclusions or discussion part, and I think it is something they should say.

• At the Model Development part, at the end, they say that they do not include in the AI the parameter GGT because it is included in the formula, but in the formula it is included in a logarithmic way, so what is really evaluated is its logarithm and not the variable itself.

• For the diagnosis of NAFLD they use the ultrasounds, here you know the diagnostic limitations of the ultrasounds, a histological diagnosis should be recommended. In spite of barriers doing that, authors should discuss this aspect.

• "The actual training sessions" section is confusing. First, they remove a number of cases because they are missing features, but they don't include how many cases were removed, so the number of cases included for training the network remained hidden. A sentence completely baffles me: "For the training phase we had operated in the same way, randomly selecting the 50 individuals extracted from the initial database", it is not clear if they have used only 50 cases to train the network (it would be very few cases), or if they refer to another stage and have named it wrong. In the AI setting, the data are divided into three groups: training (most of them), to train the network, validation (around 15%, different from the training), to evaluate the network while it is being trained, and test (at least 15% of the data, different from the other two groups), to check the functioning of the network once it is built.

• The Results part, when they analyze BRI plus Glucose etc, we can read "the algorithm accuracy percentages were compared to those of the algorithm reducing the numbers of false negatives", I don't know what it means by reducing the number of false negatives, they should to clarify it, because it could be that they compare the algorithms by seeing which has less number of false negatives or that they have eliminated false negatives to make the comparison, which in my opinion would not be correct.

• The Discussion part, they don't indicate at any time the percentage of error, and I think they should say so, because an algorithm can be better than another if one has 60% of error and the other one 70%, but in both cases they would be very big errors to consider the algorithm as good.

• References Format: many of the references are placed in different ways, before or after the point.

Reviewer #2: Dear Authors, congratulations of the innovative work to help identify NAFLD patients via machine learning.

Here are my comments:

Q1. For noninvasive diagnosis of NAFLD, proton magnetic resonance spectroscopy (H-MRS) instead of ultrasound were usually adopted for it's accuracy and less interobserver errors.

A technician or clinician performing ultrasound gives diagnosis of fatty liver when encountering higher echogenicity of liver parenchyma than that of spleen or renal cortex. The ultrasound diagnosis of steatosis is therefore subjective, prone to interobserver error, especially when studying those patients with borderline of mild fatty liver.

Did this study stratify and give different weight to a diagnosis of steatosis according to the severity noted by the ultrasound report? How did the study eliminate the interobserver error of ultrasound?

Q2. As discussed in Q1, the gold standard for diagnosis of hepatic steatosis is core biopsy, and H-MRS is regarded widely as the most accurate non-invasive diagnostic tool of steatosis. The current study will be much more persuasive if the training data set includes NAFLD confirmed by biopsy or H-MRS, and includes the ultrasound diagnosis when finally pooling the algorithms for weighting (%).

Q3. Some formatting errors or redundancy: "the accuracy of a model" in page 14. In table 2, were the numbers in the parenthesis of "Age", "FLI", "AVI", etc. indicating standard deviation? Please specify.

Q4. Are the parameters as rGT and glucose in the training models treated as continuous variables? If these were continuous variables, how did the authors describe those within normal range and those were mildly/severely deviated? The differentiation of variables within or above normal range may enhance the accuracy of the algorithm.

Q5. In figure 4, 5, 6, SVM performed nearly identical percentage of accuracy, which was relatively lower than other algorithms, while fewer wrong predictions in test phase was observed. What were the authors explanations?

Q6. As discussed in Q2, if the study aimed to "search the best algorithm to support NAFLD diagnosis", it needs a new study design. While if the study intends to show how machine learning reduce unnecessary ultrasound examinations, it only needs a minor revision.

6. PLOS authors have the option to publish the peer review history of their article (what does this mean?). If published, this will include your full peer review and any attached files.

Reviewer #1: **Yes: **Manuel Romero-Gómez

Reviewer #2: No

---

## [Author Response · Author response to Decision Letter 0]

8 Sep 2020

1) The manuscript meets PLOS ONE's style

2) We have include the full name of the ethics committee/institutional review board(s) that approved your specific study.

3) We have remove any funding-related text from the manuscript.

4) We have upload the minimal anonymized data set necessary.

Reviewer #1: Authors did a tremendous effort building the database and analyzing different AI algorithms to compare them, but I think that the networks they have built are unbalanced and some other methodological concerns that they should address.

1) The authors did make a database including 2970 patients. They used 2920 cases for the training set and only 50 to test the algorithm. From the point of view of this reviewer these 50 represent less 2% of them, but it is recommended to include at least 15% of the data for validation purpose. Moreover, cases are well represented when they are selected randomly, as they did.

1) We have modified the paragraph and hope now is more readable. The paragraph now reads. “Initially, it was important to pre-process data to eliminate missing values for features (x) or for target variables (y). Then, we obtained a dataset containing 2868 subject Later, we set the algorithm to randomly extracted 50 individuals from the initial database, which were used for the construction of a new dataset. This dataset was used for the prediction phase, making use of machine learning algorithms that had been adequately trained during the training phase. During the training sessions, all 2868 subject was used for the training and for the test by means of the 10-fold cross-validation approach (45). To retrieve the best parameters for each algorithm, GridSearchCV, which is a method contained in a Scikit-learn in Python, was used. In this way, the algorithm characterized by the lowest variance was identified, to reduce the possibility of a false prediction. After selection of the best parameters for each algorithm, we carried out the prediction on a dataset of 50 individuals (generated by the prediction), thus obtaining the predicted target variable for each algorithm.

2) Authors did use accuracy and variance to determine if the algorithm works. However, usually sensitivity and specificity are calculated because accuracy, although it takes into account the FP-FN, does not distinguish them separately, and it is as important that your algorithm is correct as it is not wrong, especially when it comes to the diagnosis of a disease.

2) We used accuracy and variance without considering specificity and sensitivity as the purpose of this study was to highlight that a carefully selected machine learning algorithm (among those considered) could considerably reduce the number of subjects to be sent for ultrasound examination and consequently also the costs of instrumental diagnosis. Furthermore, having had as output the algorithm prediction for 50 subjects for each of the models, we experimentally verified the performance of the algorithms in terms of incorrect predictions. However, as suggested by the reviewer, we evaluated the specificity and sensitivity of the SVM algorithm in the prediction phase and added the table to the text.

The table is as follows:.

Table 1 below shows the specificity and sensitivity values for each algorithm

Table 1

 Sensibility Specificity VPP VPN

FLI plus Glucose plus Age plus Sex

 0.979 1.00 1.00 0.990

AVI plus Glucose plus GGT plus Age plus Sex 0.985 1.00 1.00 0.993

BRI plus Glucose plus GGT plus Age plus

Sex 0.967 0.99 0.997 0.990

3) As for the description of the population they have used, there are several concerns. On the one hand, from the total cases, only about 32% have NAFLD, which makes the network not compensated and, be able to detect better the cases that do not have NAFLD than those that have NAFLD (something that they mention in the Discussion part, "the SVM was well able to identify subjects without NAFLD", obviously, because it has been trained with more cases without NAFLD). This can lead to very good accuracy results if the test cases are selected randomly and none of them is a NAFLD patient (and in their case it can happen since few test cases have been used). On the other hand, the network is also gender imbalanced, 56.5% male. Perhaps it would have been better to create two different networks, one for men and another for women. This decompensation is even greater in the case of women with NAFLD, there are only 30.5% of the cases, and they do not mention at any moment, neither in the conclusions or discussion part, and I think it is something they should say.

3) We have modified and added some comments in the text. We understand the concern of the reviewer and we have tried to give an explanation to the structure of our population. The paragraph now reads: “Data come from a population-based random sample of subjects. As the sampling procedure was random, the sample represents not only the age-sex structure of the population but also the age-sex distribution of diseases or conditions present in the population. Then, the frequency of NAFLD represents the prevalence of NAFLD in the population. Moreover, in the population studied NAFLD is a condition more prevalent among men than in women.”

4) At the Model Development part, at the end, they say that they do not include in the AI the parameter GGT because it is included in the formula, but in the formula it is included in a logarithmic way, so what is really evaluated is its logarithm and not the variable itself.

4) Yes. In the formula the natural logarithm of GGT is evaluated. Authors provided an explanation in the original paper (Bedogni G et al. The Fatty Liver Index: a simple and accurate predictor of hepatic steatosis in the general population. BMC Gastroenterol 2006; 6:33: “All variables besides gender were evaluated as continuous predictors. Linearity of logits was ascertained using the Box-Tidwell procedure [22]. To obtain a linear logit, we transformed age using the coefficient suggested by the Box-Tidwell procedure [(age/10) ∧ 4.9255] and ALT, AST, GGT, insulin and triglycerides using natural logarithms (loge). The logits of the other predictors (BMI, waist circumference, glucose, cholesterol, ethanol and the sum of 4 skinfolds) were linear.” Tomake it clearer we have added a comment in the text. Now it reads: “

5) For the diagnosis of NAFLD they use the ultrasounds, here you know the diagnostic limitations of the ultrasounds, a histological diagnosis should be recommended. In spite of barriers doing that, authors should discuss this aspect.

5) We have added in text, wich now reads: “. Limitations include both the limited number of observations and the method used to perform NAFLD diagnosis. It may be criticized the low sensibility of the NAFLD diagnostic methodology, as it fails to detect fatty liver content >25-90% (47). However, this is a population-based study and subjects were chosen from the electoral register. They did not seek medical attention and participated on volunteer basis. Then, the diagnosis of NAFLD performed by US was the only diagnostic procedure we could propose to the participants. Ethical issues prevent us to propose biopsy or H-MRS. Moreover, and to lighten the waiting lists, our purpose was to find out a machine learning algorithm that permit us to avoid a number of USs which otherwise would have been prescribed. Then, this algorithm is useful to exclude NAFLD and as valid support diagnostic in the context of epidemiologic studies and not as replacement diagnostic tool.” 

6) "The actual training sessions" section is confusing. First, they remove a number of cases because they are missing features, but they don't include how many cases were removed, so the number of cases included for training the network remained hidden. A sentence completely baffles me: "For the training phase we had operated in the same way, randomly selecting the 50 individuals extracted from the initial database", it is not clear if they have used only 50 cases to train the network (it would be very few cases), or if they refer to another stage and have named it wrong. In the AI setting, the data are divided into three groups: training (most of them), to train the network, validation (around 15%, different from the training), to evaluate the network while it is being trained, and test (at least 15% of the data, different from the other two groups), to check the functioning of the network once it is built.

6) "The actual training sessions" section has been modified and now reads: “ Initially, it was important to pre-process data to eliminate missing values for features (x) or for target variables (y). Then, we obtained a dataset containing 2868 subject Later, we set the algorithm to randomly extracted 50 individuals from the initial database, which were used for the construction of a new dataset. This dataset was used for the prediction phase, making use of machine learning algorithms that had been adequately trained during the training phase. During the training sessions, all 2868 subject was used for the training and for the test by means of the 10-fold cross-validation approach (45). To retrieve the best parameters for each algorithm, GridSearchCV, which is a method contained in a Scikit-learn in Python, was used. In this way, the algorithm characterized by the lowest variance was identified, to reduce the possibility of a false prediction. After selection of the best parameters for each algorithm, we carried out the prediction on a dataset of 50 individuals (generated by the prediction), thus obtaining the predicted target variable for each algorithm.”

7) The Results part, when they analyze BRI plus Glucose etc, we can read "the algorithm accuracy percentages were compared to those of the algorithm reducing the numbers of false negatives", I don't know what it means by reducing the number of false negatives, they should to clarify it, because it could be that they compare the algorithms by seeing which has less number of false negatives or that they have eliminated false negatives to make the comparison, which in my opinion would not be correct.

7) We thank the reviewer. The sentence was incorrect. We have cancelled it out.

8) The Discussion part, they don't indicate at any time the percentage of error, and I think they should say so, because an algorithm can be better than another if one has 60% of error and the other one 70%, but in both cases they would be very big errors to consider the algorithm as good.

We have added in text, wich now reads: “.We obtained in the test phase for the models composed of FLI plus Glucose plus Age plus Sex and AVI plus Glucose plus GGT plus Age plus Sex a percentage error equal to 32% while for the model composed of BRI plus Glucose plus GGT plus Age plus Sex an error of 23%. However, in the prediction phase, the model that made fewer errors was the one composed of AVI plus Glucose plus GGT plus Age plus Sex with an error of 20 % while FLI plus Glucose plus Age plus Sex 26% and BRI plus Glucose plus GGT plus Age plus Sex 28%. Therefore, AVI plus Glucose plus GGT plus Sex plus Age was the model that contributed most to reducing unnecessary ultrasound examinations.”

9) References Format: many of the references are placed in different ways, before or after the point. 

9) The references have been corrected and all inserted before the point.

Reviewer #2: Dear Authors, congratulations of the innovative work to help identify NAFLD patients via machine learning.

Here are my comments:

Q1. For noninvasive diagnosis of NAFLD, proton magnetic resonance spectroscopy (H-MRS) instead of ultrasound were usually adopted for it's accuracy and less interobserver errors.

A technician or clinician performing ultrasound gives diagnosis of fatty liver when encountering higher echogenicity of liver parenchyma than that of spleen or renal cortex. The ultrasound diagnosis of steatosis is therefore subjective, prone to interobserver error, especially when studying those patients with borderline of mild fatty liver.

Did this study stratify and give different weight to a diagnosis of steatosis according to the severity noted by the ultrasound report? How did the study eliminate the interobserver error of ultrasound?

Q2. As discussed in Q1, the gold standard for diagnosis of hepatic steatosis is core biopsy, and H-MRS is regarded widely as the most accurate non-invasive diagnostic tool of steatosis. The current study will be much more persuasive if the training data set includes NAFLD confirmed by biopsy or H-MRS, and includes the ultrasound diagnosis when finally pooling the algorithms for weighting (%).

A1, A2. We have explained and added a paragraph in the text. Now it reads: “Following the European Association for the Study of the Liver (EASL), European Association for the Study of Diabetes (EASD) and European Association for the Study of Obesity (EASO) recommendations (7), NAFLD diagnosis was performed using an ultrasound scanner Hitachi H21 Vision (Hitachi Medical Corporation, Tokyo, Japan). Examination of the visible liver parenchyma was performed with a 3.5 MHz transducer.” And “It may be criticized the low sensibility of the NAFLD diagnostic methodology, as it fails to detect fatty liver content >25-90% (47). However, this is a population-based study and subjects were chosen from the electoral register. They did not seek medical attention and participated on volunteer basis. Then, the diagnosis of NAFLD performed by US was the only diagnostic procedure we could propose to the participants. Ethical issues prevent us to propose biopsy or H-MRS. Moreover, and to lighten the waiting lists, our purpose was to find out a machine learning algorithm that permit us to avoid a number of USs which otherwise would have been prescribed. Then, this algorithm is useful to exclude NAFLD and as valid support diagnostic in the context of epidemiologic studies and not as replacement diagnostic tool.”

Q3. Some formatting errors or redundancy: "the accuracy of a model" in page 14. In table 2, were the numbers in the parenthesis of "Age", "FLI", "AVI", etc. indicating standard deviation? Please specify.

A3 We have modified the Table 2. We have added a footnote that reads: Cells reporting subject characteristics contain mean (±SD) or n (%).

Q4. Are the parameters as GGT and glucose in the training models treated as continuous variables? If these were continuous variables, how did the authors describe those within normal range and those were mildly/severely deviated? The differentiation of variables within or above normal range may enhance the accuracy of the algorithm.

A4. We have considered the indexes in the original form they have been published after a peer review process and without any type of modification. Anyway, we have added a paragraph about this in the text. The text now reads: “We choose to introduce the biochemical markers as continuous variables to better reflect the natural scale of the variables. The assumption behind this choice is that the effect of categorization is the loss of information”.

Q5. In figure 4, 5, 6, SVM performed nearly identical percentage of accuracy, which was relatively lower than other algorithms, while fewer wrong predictions in test phase was observed. What were the authors explanations?

A5 The lowest number of incorrect predictions in the test phase is due to the fact that even if SVM performed with lower accuracy than the others it was the one with the least variance therefore in the test phase it was less wrong than any other algorithm with a more accuracy percentage but characterized by a larger variance.

Q6. As discussed in Q2, if the study aimed to "search the best algorithm to support NAFLD diagnosis", it needs a new study design. While if the study intends to show how machine learning reduce unnecessary ultrasound examinations, it only needs a minor revision. 

A6 (The article wanted to highlight once the best algorithm capable of supporting the diagnosis of NAFLD among the algorithms studied was selected, how much such an approach could affect the reduction of costs and the number of unnecessary ultrasound examinations)

Moreover and, to make it clearer our methods we have added the following paragraph to the text: “A fasting venous blood sample was drawn, and the serum was separated into two different aliquots. One aliquot was immediately stored at −80 °C. The second aliquot was used to test biochemical serum markers by standard laboratory techniques in our Central laboratory.”

---

## [Decision Letter · Decision Letter 1]

5 Oct 2020

SELECTING THE BEST MACHINE LEARNING ALGORITHM TO SUPPORT THE DIAGNOSIS OF NON-ALCOHOLIC FATTY LIVER DISEASE: A META LEARNER STUDY.

PONE-D-20-19706R1

Dear Dr. Osella,

We’re pleased to inform you that your manuscript has been judged scientifically suitable for publication and will be formally accepted for publication once it meets all outstanding technical requirements.

Kind regards,

Ming-Lung Yu, MD, PhD

Academic Editor

PLOS ONE

Additional Editor Comments (optional):

Reviewers' comments:

Reviewer's Responses to Questions

**Comments to the Author**

1. If the authors have adequately addressed your comments raised in a previous round of review and you feel that this manuscript is now acceptable for publication, you may indicate that here to bypass the “Comments to the Author” section, enter your conflict of interest statement in the “Confidential to Editor” section, and submit your "Accept" recommendation.

Reviewer #2: All comments have been addressed

2. Is the manuscript technically sound, and do the data support the conclusions?

Reviewer #2: Partly

3. Has the statistical analysis been performed appropriately and rigorously? 

Reviewer #2: Yes

4. Have the authors made all data underlying the findings in their manuscript fully available?

Reviewer #2: Yes

5. Is the manuscript presented in an intelligible fashion and written in standard English?

Reviewer #2: Yes

6. Review Comments to the Author

Reviewer #2: The authors have already addressed all of the concerns I raised.

The paper showed machine learning approach could be an efficient and low cost tool to support diagnosis of NAFLD.

7. PLOS authors have the option to publish the peer review history of their article (what does this mean?). If published, this will include your full peer review and any attached files.

Reviewer #2: No

---

## [Editor Report · Acceptance letter]

9 Oct 2020

PONE-D-20-19706R1 

Selecting the Best Machine Learning Algorithm to support the diagnosis of Non-Alcoholic Fatty Liver Disease: A Meta Learner Study. 

Dear Dr. Osella:

I'm pleased to inform you that your manuscript has been deemed suitable for publication in PLOS ONE. Congratulations! Your manuscript is now with our production department. 

Kind regards, 

on behalf of

Dr. Ming-Lung Yu 

Academic Editor

PLOS ONE